# Synthesis of LiDAR-Detectable True Black Core/Shell Nanomaterial and Its Practical Use in LiDAR Applications

**DOI:** 10.3390/nano12203689

**Published:** 2022-10-20

**Authors:** Suk Jekal, Jiwon Kim, Dong-Hyun Kim, Jungchul Noh, Min-Jeong Kim, Ha-Yeong Kim, Min-Sang Kim, Won-Chun Oh, Chang-Min Yoon

**Affiliations:** 1Department of Chemical and Biological Engineering, Hanbat National University, Yuseong-gu, Daejeon 34158, Korea; 2McKetta Department of Chemical Engineering and Texas Material Institute, The University of Texas at Austin, Austin, TX 78712, USA; 3Department of Advanced Materials Science and Engineering, Hanseo University, Seosan-si 31962, Korea

**Keywords:** LiDAR-detectable, LiDAR black, NIR-reflective, autonomous vehicle, core/shell, blackness, nanomaterial

## Abstract

Light detection and ranging (LiDAR) sensors utilize a near-infrared (NIR) laser with a wavelength of 905 nm. However, LiDAR sensors have weakness in detecting black or dark-tone materials with light-absorbing properties. In this study, SiO_2_/black TiO_2_ core/shell nanoparticles (SBT CSNs) were designed as LiDAR-detectable black materials. The SBT CSNs, with sizes of 140, 170, and 200 nm, were fabricated by a series of Stöber, TTIP sol-gel, and modified NaBH_4_ reduction methods. These SBT CSNs are detectable by a LiDAR sensor and, owing to their core/shell structure with intrapores on the shell (*ca*. 2–6 nm), they can effectively function as both color and NIR-reflective materials. Moreover, the LiDAR-detectable SBT CSNs exhibited high NIR reflectance (28.2 R%) in a monolayer system and true blackness (*L** < 20), along with ecofriendliness and hydrophilicity, making them highly suitable for use in autonomous vehicles.

## 1. Introduction

Autonomous vehicles (i.e., self-driving cars) are the technology that has received some of the most attention in the smart mobility industry, along with electric vehicles. Autonomous vehicles should be able to recognize the surrounding environment and thus avoid accidents and drive safely to their destination without human manipulation [1,2]. To implement a completely autonomous driving system, various sensor systems, including radars, cameras, and light detection and ranging (LiDAR) sensors, which act as the eyes of the autonomous vehicle, have been extensively studied by various research groups and companies [3]. Among these sensors, LiDAR has received widespread attention owing to its high resolution, accuracy, and ability to generate three-dimensional maps, as well as the fact that it is not as affected by rain and fog, in contrast to radar and cameras [4].

The principle of LiDAR sensors is that they emit a near-infrared (NIR) laser with a wavelength of approximately 905 nm and detect its time of reflection [5,6]. Owing to the high degree of straightness of the laser, LiDAR sensors have a higher subject recognition efficiency, which depends on both the distance and width of the object being detected. Hence, significant efforts have been devoted by major automotive electronics companies to improve LiDAR devices. However, despite the significant improvements, LiDAR sensors have a fundamental drawback related to the detection of black and dark-tone materials. Because black and dark-tone materials absorb light, including that in the NIR region, the precision and accuracy of LiDAR sensors decrease significantly during the detection of these materials [7]. Humans have strong color preferences, and dark-toned objects make up a large proportion of vehicle components, human outfits, and industry markets [8]. Therefore, it is essential to overcome this drawback of LiDAR sensor systems for the realization of perfect autonomous vehicles and environments.

Recently, LiDAR-detectable (i.e., NIR-reflective) dark-tone materials have received wide attention. There are several ways of incorporating NIR-reflective layers on dark-tone-colored vehicles, including using inorganic pigments, organic dark-tone perylene-derived pigments, and mixed pigments (Figure 1). In the case of inorganic pigments, various metal oxides, including Mg-doped ZnFeO_4_, Al_2_O_3_-doped CoFe_2_O_4_, and Ce_1−x_Gd_x_VO_4_ pigments, exhibit NIR reflectances of ~25–70 R% [9,10,11]. However, inorganic pigments have two drawbacks; namely, they show high *L** values and metal toxicity. Specifically, a lower *L** value indicates higher material blackness. The *L** values of inorganic pigments that have been reported in previous studies are approximately 23–49 and closer to those of dark-tone gray and brown materials instead of those of black ones [9,10,11]. On the other hand, perylene-derived organic materials can also be employed as LiDAR-detectable pigments. However, these organic pigments require a complex bilayered system of stacked dark-tone colors and a LiDAR-reflective layer of a TiO_2_-based material [12,13]. The fabrication of a bilayered system would involve additional coating processes, making it more expensive and time-consuming. Moreover, perylene-derived organic materials have additional disadvantages as well, such as a low degradation temperature and high toxicity. Accordingly, there is an urgent need to develop new materials to overcome these limitations of conventional LiDAR-detectable pigments and coating systems.

Titanium dioxide (TiO_2_) is one of the most effective NIR-reflective materials in the 780–2500 nm range due to its white color [14]. In addition, TiO_2_ can be readily incorporated into other metal oxides, such as silica (SiO_2_), using a typical sol-gel method. Numerous studies have reported the synthesis of SiO_2_/TiO_2_ core/shell materials with desirable properties in terms of the specific surface area, pore volume, hydrophilicity, and toxicity [15,16,17]. SiO_2_/TiO_2_ core/shell materials are employed in numerous applications, including as light-scattering materials for dye-sensitized solar cells, nanocarriers for drug delivery, and photocatalysts [18,19,20,21,22,23]. Furthermore, the properties of the outermost TiO_2_ shell can be manipulated through various surface treatments [24]. For instance, noble metals, such as Au and Pt can be introduced into the TiO_2_ shell via silylation [25,26]. In addition, various alkaline earth metals can be doped onto the TiO_2_ shell using the sonication-mediated etching method [27,28]. In addition, the TiO_2_ shell can be turned black by the NaBH_4_ reduction method. However, the synthesis of SiO_2_/black TiO_2_ core/shell materials has not been reported previously.

In this paper, we report the synthesis of newly designed SiO_2_/black TiO_2_ core/shell nanoparticles (SBT CSNs), which are a highly LiDAR-detectable material. The SBT CSNs were synthesized using a combination of the sol-gel and NaBH_4_ reduction methods. The synthesized SBT CSNs exhibited very high NIR reflectances, which are approximately 28.2% and 50.2 R% in the monolayer (without a TiO_2_ surfacer) and bilayer (with TiO_2_ surfacer) forms, respectively. Furthermore, SBT CSN-coated black objects were detected using an actual LiDAR sensor, which could readily perceive the objects. To the best of our knowledge, this is the first study to report a method for fabricating true black (*L** < 20) LiDAR-detectable materials without using any organic perylene-derived materials or inorganic toxic metals. In this regard, this study opens new possibilities for the development of LiDAR-reflective materials that are both efficient and ecofriendly, and thus, in line with the goals of the smart mobility industry.

## 2. Materials and Methods

### 2.1. Materials

Tetraethyl orthosilicate (TEOS, 98%) and titanium (IV) isopropoxide (TTIP, 97%) were purchased from Sigma-Aldrich Chemical Co. (Burlington, MA, USA). An ammonia solution (NH_4_OH, 28.0%), ethyl alcohol (EtOH, 99.9%), acetonitrile (99.8%), sodium borohydride (NaBH_4_, 98.0%), acetylene carbon black (99.99%), and 250 nm-sized anatase phase titanium dioxide powder (TiO_2_, 98.5%) were purchased from Samchun Chemical Co. (Seoul, Korea). All the chemicals, reagents, and solvents were used as received without further purification.

### 2.2. Synthesis of SiO_2_/TiO_2_ Core/Shell Nanoparticles (ST CSNs)

A SiO_2_ colloidal suspension was prepared using the typical Stöber method. Briefly, ethyl alcohol (101 mL), deionized water (2.1 mL), and the ammonia solution (5.2 mL) were mixed under vigorous magnetic stirring for 15 min. Subsequently, TEOS (3.3 mL) was added to the as-prepared solution, and the reaction was allowed to proceed for 6 h to obtain a SiO_2_ colloidal solution. The ST CSNs were synthesized based on the typical sol-gel reaction of metal alkoxides. A mixture of ethyl alcohol (20 mL), acetonitrile (10 mL), and TTIP (5 mL) was slowly added to the as-synthesized SiO_2_ colloidal solution while maintaining the temperature at 4 °C. The sol-gel reaction was allowed to continue for 12 h under vigorous stirring. The synthesized ST CSNs were collected by centrifugation at 12,000 rpm for 30 min, and the impurities were removed by three additional washes and centrifugation. Finally, the washed ST CSNs were dried overnight in an oven (90 °C). To control the size of the SBT CSNs, the starting amount of the TEOS was controlled at 3.3, 3.8, and 4.2 mL, resulting in SiO_2_ core templates with sizes of 100, 120, and 140 nm, respectively.

### 2.3. Fabrication of SiO_2_/Black TiO_2_ Core/Shell Black Nanoparticles (LiDAR-Detectable SBT CSNs)

The SBT CSNs were prepared using a modified NaBH_4_ reduction method. Typically, the as-synthesized ST CSNs (0.5 g) and NaBH_4_ (0.3 g) were mixed thoroughly using a mortar and pestle. The mixed powder was carefully transferred into an alumina boat, which was placed in a tube furnace and heated for 40 min at 550 °C at a heating rate of 5 °C/min in a nitrogen atmosphere. After the heat treatment, the SBT CSNs were washed by being dispersed in hot deionized water to remove the organic and inorganic residues and centrifuged three times. Finally, the washed SBT CSNs were dried overnight in an oven (90 °C).

### 2.4. Characterization

The morphologies of the ST CSNs and SBT CSNs were analyzed using field-emission scanning electron microscopy (FE-SEM, SU8230, Hitachi, Tokyo, Japan) and transmission electron microscopy (TEM, HF5000, Hitachi, Tokyo, Japan). The specific surface areas and pore volumes of the materials were evaluated based on their N_2_-sorption curves (TriStar II 3020, Micromeritics, Norcross, GA, USA) using the Brunauer–Emmett–Teller (BET) method. STEM elemental mapping images were acquired by a TEM equipped with a scanning transmission electron microscope (STEM, HD-2700, Hitachi, Tokyo, Japan). The elemental compositions of the materials were detected by energy-dispersive X-ray spectroscopy (EDS, Ultim Max, Oxford Instruments, Abingdon, UK) installed on the TEM. The surface chemical and oxidation states of the materials were investigated using X-ray photoelectron spectroscopy (XPS, K-alpha, Thermo Fisher Scientific, Waltham, MA, USA). The crystalline structures of the materials were examined using X-ray diffraction analysis (XRD, D8 Advance, Bruker Co., Billerica, MA, USA). The NIR reflectances of the materials were measured using a diffuse reflectance UV-VIS-NIR spectrophotometer (SolidSpec-3700, Shimadzu, Kyoto, Japan).

### 2.5. Measurement of Blackness of SBT CSN-Based Paint

The blackness and color of the SBT CSNs were measured with a colorimeter (3nh, NR60CP) using the CIE *L*a*b** system. To measure the blackness and color, the SBT CSNs were mixed with clear varnish (Jevisco, W21602) at a ratio of 7:1 under vigorous magnetic stirring for 1 h and then subjected to ultrasonication for 20 min. Next, the well-dispersed SBT CSN-based paint was left to stand for 5 min until it stabilized. The paint was then sprayed onto a glass substrate (5 × 5 cm^2^ with a thickness of 0.3 cm) using a spray gun (Daiwa, DSP-090) to form a coating with a thickness of approximately 30 μm. For comparison, black, gray, and brown paints were also sprayed on glass substrates using the same experimental procedure. The sprayed samples were dried at room temperature (25 °C), and their brightness and color were measured five times using a colorimeter for accuracy. The average *L*a*b** values of each sample were determined.

### 2.6. Measurement of NIR Reflectance and Evaluation of LiDAR Detectability of SBT CSN-Based Paint

To investigate the NIR reflectance of the SBT CSNs, the CSNs were prepared under two different conditions: without a TiO_2_ surfacer and with a TiO_2_ surfacer. The formation of a bilayer with a TiO_2_ white surfacer is the conventional method for ensuring high NIR reflectance in the case of organic perylene-based dark-tone LiDAR-reflective coating systems. The samples used for the NIR reflectance measurements and LiDAR detectability tests were prepared using the same procedure as that for the color measurements. Specifically, various-sized SBT CSNs, a black pigment (carbon black), and TiO_2_ were mixed thoroughly in clear varnish at a ratio of 7:1. The well-mixed paints were then sprayed onto glass substrates to form coatings with a thickness of approximately 30 μm using a spray gun and dried at room temperature. To prepare the TiO_2_-surfacer samples, commercially obtained TiO_2_ was mixed with clear varnish and sprayed onto glass substrates. After the coatings had dried, the SBT CSN-based paints were sprayed onto the white TiO_2_-coated glass substrates to form the bilayered samples. The LiDAR detectabilities of the samples were tested using a LiDAR sensor with a detection range of approximately 905 nm (Intel RealSense L515). The SBT CSN-based painted monolayered samples and black pigment-coated substrates were placed on black paper, and a LiDAR sensor was used in the depth mode to detect the samples. The output was viewed on a screen.

## 3. Results and Discussion

### 3.1. Characterization of SBT CSNs

The strategy employed in this study to develop a new black, LiDAR-detectable material was to design a material that is free of a bilayer, such as that used in the case of conventional perylene-based black pigments, as well as harmful heavy metals but that still exhibits high NIR reflectance and ecofriendliness. The NIR reflectance mechanisms of the conventional dark-tone organic/inorganic pigment-based materials and the SBT CSNs synthesized in this study are shown in Figure 2.

A schematic diagram of the method used for synthesizing the SBT CSNs is shown in Figure 3. First, a colloidal suspension of uniformly sized SiO_2_ particles was formed by sol-gel condensation for use as the template material. TiO_2_ shells were continuously formed on the surfaces of the SiO_2_ nanoparticles. Owing to the low reaction temperature, the TTIP precursor was slowly converted into TiO_2_, resulting in the formation of a porous structure on the SiO_2_ nanoparticles. This yielded the ST CSNs. Finally, using the NaBH_4_ reduction method, the TiO_2_ shells on the ST CSNs were successfully turned black to obtain the SBT CSNs.

The morphologies of the SiO_2_ nanoparticles, ST CSNs, and SBT CSNs were analyzed using FE-SEM to confirm the successful and uniform synthesis of the materials, as shown in Figure 4. The diameters of the SiO_2_ nanoparticles, ST CSNs, and SBT CSNs were determined to be approximately 100, 130, and 140 nm, respectively. Moreover, SBT CSNs with sizes of 170 and 200 nm and their precursor materials (SiO_2_ nanoparticles with sizes of 120 and 140 nm, respectively, and ST CSNs with sizes of 160 and 190 nm, respectively) were also successfully synthesized using the same procedure; the only difference was in the size of the SiO_2_ core templates used. The core/shell structures of the ST CSNs and SBT CSNs were analyzed using TEM, as shown in Figure 5. It is clear that all the ST CSNs, as well as the 140, 170, and 200 nm SBT CSNs, maintained their core/shell structure. In addition, small intrapores were observed on the outer TiO_2_ shells, confirming the porous nature of the ST CSNs and SBT CSNs.

To further characterize the porous nature of the materials, their BET surface areas and Barrett–Joyner–Halenda (BJH) pore distributions were calculated from their N_2_-sorption isotherms, as shown in Figure 6. The SiO_2_ nanoparticles showed a typical type II isotherm corresponding to the nonporous materials [29]. On the other hand, the ST CSNs and SBT CSNs exhibited typical type IV hysteresis isotherms corresponding to porous materials [30]. The pore sizes and BET specific surface areas of the SiO_2_ nanoparticles, ST CSNs, and SBT CSNs are listed in Table 1. The specific surface area of the materials increased with the formation of the TiO_2_ shells. For instance, the 100 nm SiO_2_ nanoparticles had a specific surface area of 74.1 m^2^∙g^−1^. However, after the formation of the shells, the specific surface area increased dramatically, to 262.2 m^2^∙g^−1^ in the case of the 130 nm ST CSNs. This trend in the specific surface area was observed for all the ST CSNs. In addition, the smaller-sized materials also showed higher specific surface areas, which was in accordance with the nature of nanomaterials [31]. After the NaBH_4_ heat reduction process, the specific surface areas of the SBT CSNs were slightly lower compared to those of the precursor ST CSNs. This phenomenon is attributable to a change in the crystallinity of the TiO_2_ shells, which were amorphous in the case of the ST CSNs and anatase in the case of the SBT CSNs. However, the 140 nm SBT CSNs maintained a high specific surface area of 208.8 m^2^∙g^−1^ even after the reduction process. Moreover, uniform intrapores with sizes of approximately 2.8–2.9 nm were formed in the TiO_2_ shells of the ST CSNs. Interestingly, the SBT CSNs contained pores 2.5–2.6 nm and 6.7–7.1 nm in size. These newly formed slightly larger pores were the result of a change in the crystalline phase to anatase, as mentioned previously [32,33]. Accordingly, the SBT CSNs exhibited a core/shell structure with high porosity and specific surface area owing to the formation of the porous TiO_2_ shells on the SiO_2_ core templates. Various studies have reported that core/shell materials with porous shells exhibit outstanding light scattering and reflectance characteristics [34]. Therefore, the proposed approach for synthesizing black materials using a porous core/shell structure is advantageous with respect to improving the NIR reflectance and hence LiDAR detectability.

The crystallinities and phases of the SiO_2_ nanoparticles, ST CSNs, and SBT CSNs were investigated using XRD analysis (Figure 7). The SiO_2_ nanoparticles and ST CSNs displayed a broad peak at 20–30°, indicating that these materials were amorphous, owing to the sol-gel process. In contrast, the SBT CSNs exhibited sharp peaks related to the (101), (004), (200), (105), (204), (116), (215), and (303) planes, which matched perfectly the XRD pattern of anatase TiO_2_ (JCPDS-21-1272) [35]. Thus, the XRD pattern of the SBT CSNs confirmed the phase change of the TiO_2_ shells from amorphous in the case of the ST CSNs to anatase in the case of the SBT CSNs after the NaBH_4_ reduction process at 550 °C.

XPS was performed to investigate the chemical and molecular characteristics of the SiO_2_ nanoparticles, ST CSNs, and SBT CSNs, as shown in Figure 8. The Si 2p spectra of the SiO_2_ nanoparticles, ST CSNs, and SBT CSNs confirmed the presence of SiO_2_ as the core material [36]. In the case of the ST CSNs and SBT CSNs, additional Ti 2p signals were also observed, confirming the successful formation of TiO_2_ shells on the SiO_2_ cores. Note that Ti 2p signals were detected at approximately 463.7 and 458.0 eV in the case of the ST CSNs, indicating the presence of Ti^4+^ species of pure TiO_2_ [37]. On the other hand, an additional Ti 2p peak was observed at 454.2 eV in the case of the SBT CSNs, which was attributed to the Ti^3+^ species [38]. A previous study reported that the Ti^4+^ species in TiO_2_ was reduced to Ti^3+^ during NaBH_4_ reduction [39]. Furthermore, the reduction of the Ti species was observed in the case of all the SBT CSNs, and the Ti^3+^/Ti^4+^ ratios, as calculated from the XPS data, are listed in Table 2.

Furthermore, scanning transmission electron microscopy (STEM) elemental mapping analysis was conducted to visualize and confirm the elemental and structural composition of the as-synthesized 100 nm SiO_2_ nanoparticles, 130 nm ST CSNs, and 140 nm SBT CSNs (Figure 9). It was clear that the SiO_2_ nanoparticles were only composed of Si and O elements. For the ST CSNs and SBT CSNs, additional Ti elements were detected along with Si and O, indicating the successful formation of TiO_2_ shells on the core SiO_2_ nanoparticles via the sol-gel method. Moreover, it was evident that the Si, Ti, and O elements were clearly presented for the SBT CSNs, indicating that the elemental compositions of materials did not change even after the NaBH_4_ reduction. The analysis result of the energy-dispersive X-ray spectroscopy (EDS) further quantified the elemental composition (Appendix A). Similar to STEM, the SiO_2_ nanoparticles only contained Si and O. In contrast, the ST CSNs and SBT CSNs were detected with Si, Ti, and O elements with similar Ti/Si ratios of 0.36 and 0.31. The XPS results confirm that some Ti^4+^ species of the TiO_2_ shells were reduced to Ti^3+^ species owing to the NaBH_4_ reduction, but the overall Ti elemental composition did not change after the reduction process, as confirmed by the EDS quantification analysis.

### 3.2. Blackness and Color of LiDAR-Detectable SBT CSNs

The color of the SBT CSNs was measured using a colorimeter. In general, the color of a material can be determined using the CIE *L*a*b** system. The *L** value represents the lightness of the material in the 0–100 range, with lower *L** values meaning the material is blacker. In addition, *a** represents the red (positive value) and green (negative value) color states, while *b** represents the yellow (positive value) and blue (negative value) color states. To measure the blackness and color of the SBT CSNs, a paint of the SBT CSNs was produced by mixing them in a clear hydrophilic varnish. Owing to the hydrophilicity of the varnish, the SBT CSNs were readily dispersed in it and formed a hydrophilic paint (i.e., a water-based paint). It is known that hydrophilic paints have several advantages, such as a low VOC content, ease of application, and fast drying [40]. In addition, water-based paints are regarded as suitable materials for smart mobility applications and devices, including autonomous and electric vehicles, which place an emphasis on ecofriendliness [41]. For testing, the SBT CSN-based paint was sprayed onto a glass substrate. For comparison, paints of other colors (black, brown, and gray) were also coated on glass substrates using the same experimental procedure. Note that the black paint exhibited an *L** value of 10.9 and *a** and *b** values of 0.8 and −2.2, respectively. Importantly, the SBT CSN-based paint showed an *L** value of 11.4, which was similar to that of the black paint, indicating that the SBT CSNs were true black (Figure 10). Generally, true black color in the CIE *L*a*b* system exhibits *L** values lower than 20 and *a** and *b** values of −3 to 3 [42,43]. With respect to the brown and gray paints, their *L**, *a**, and *b** values did not match those of the black paint. Thus, it was confirmed that the SBT CSNs were true black in color. As mentioned previously, the effective reduction of the TiO_2_ shells resulted in a change in the color of the CSNs to black to the human eye under visible light.

### 3.3. NIR Reflectance and Practical Use of SBT CSNs in LiDAR Applications

To investigate the NIR reflectances of the various SBT CSN samples, substrates coated with paints based on the samples were analyzed using a UV-VIS-NIR spectrophotometer. Generally, most commercial LiDAR sensors utilize NIR wavelengths near 905 nm to lower the cost [44]. Single coatings of the paints formed using the 140, 170, and 200 nm SBT CSNs exhibited NIR reflectances of 28.2, 25.2, and 22.1 R%, respectively, at 905 nm without an NIR-reflective additive, such as a TiO_2_ surfacer or white material (Figure 11a). Detailed NIR reflectances of SBT CSNs in monolayer and bilayer are listed in Table 3. The commercially obtained carbon black exhibited a low NIR reflectance of 5.4 R% owing to its light-absorbing nature. In addition, the NIR reflectances of the SBT CSNs increased with a decrease in their size. This result is in accordance with the specific surface areas of the materials, which are listed in Table 1. Owing to the high specific surface area of the CSNs, the 140 nm SBT CSN-based paint showed the highest R% value. The NIR reflectance of the SBT CSNs can be attributed to the synergistic effects of their high specific surface area and porous nature and the difference in the refractive indices of the SiO_2_ core and TiO_2_ shell, which resulted in light reflection and scattering in the NIR region [18,45]. Previous studies have reported that the light reflectivity of core/shell or bilayered materials increases with an increase in the difference in the refractive indices of the constituent materials. Moreover, the NIR reflectance of the SBT CSN-based paints was also investigated using a bilayered system, as shown in Figure 11b. With the use of a TiO_2_ surfacer, the NIR reflectance of the SBT CSNs increased to 50.2 R%. This can be attributed to the additional reflection of the NIR light escaping from the SBT CSNs by the TiO_2_ surfacer in the bilayered system. The newly designed SBT CSNs were compared with other dark-tone LiDAR-detectable materials are summarized in Table 4. Compared to the previously reported dark-tone LiDAR-detectable materials, it was clearly observed that the SBT CSNs were synthesized with true blackness (*L** of 11.4), high NIR reflectance in the monolayer (R% of 28.2), and no heavy metal inclusion. With these various advantages, SBT CSNs can be a promising candidate for application in future autonomous vehicles.

Finally, the practical LiDAR sensor application of the SBT CSNs was evaluated by employing a monolayer of the 140 nm SBT CSN-based and black paints, as shown in Figure 12. It is evident that the SBT CSN-based paint was recognized by the LiDAR sensor and visualized on the screen (the LiDAR sensor was used in the depth mode). Thus, the SBT CSNs synthesized in this study can serve as both color- and NIR-reflective materials without the assistance of white materials. Furthermore, their advantages of ecofriendliness, hydrophilicity, and nontoxicity make them suitable for use as next-generation materials for smart mobility and autonomous vehicle applications.

## 4. Conclusions

In conclusion, LiDAR-detectable SBT CSNs were successfully synthesized by the Stöber, TTIP sol-gel, and modified NaBH_4_ reduction methods, which were performed in sequence. Each experimental step resulted in the following beneficial effects: the TiO_2_ shell coating increased the specific surface area, while the reduction process turned the CSNs black. Of the various-sized SBT CSNs fabricated, the 140 nm SBT CSNs exhibited the highest specific surface area, which provided a beneficial effect on NIR reflectance. To measure the NIR reflectances, paints were prepared by mixing the SBT CSNs with commercially obtained clear varnish, and these paints were sprayed on glass substrates. The paint based on the 140 nm SBT CSNs exhibited an NIR reflectance of 28.2 R% at a wavelength of 905 nm without the use of an NIR-reflective layer. The high NIR reflectance of the SBT CSNs suggests that they can be used both as color- and NIR-reflective materials. Specifically, the high specific surface area and intrapores of the SBT CSNs and the difference in the refractive indices of their core and shell materials are responsible for their high NIR reflectance. Moreover, a bilayered system consisting of a paint based on the SBT CSNs and a TiO_2_ surfacer manifested an even higher NIR reflectance at 50.2 R%, owing to the additional reflectance induced by the escaped NIR light. Finally, the SBT CSN-based paint could be detected with ease by a commercially available LiDAR sensor. Thus, the LiDAR-detectable black SBT CSNs synthesized in this study, which are also ecofriendly and hydrophilic, should aid in the design and development of new types of materials for autonomous vehicles in the future.

## Figures and Tables

**Figure 1 nanomaterials-12-03689-f001:**
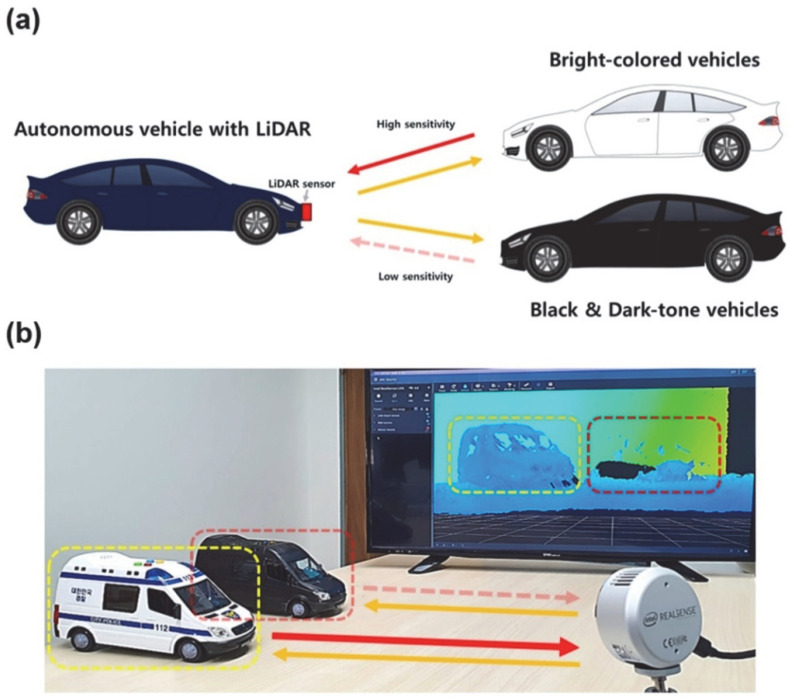
(**a**) Conceptual illustration of NIR reflectivity of white- and black-colored cars and their low- and high-sensitivities on LiDAR sensors. (**b**) Digital photograph of white- and black-colored car objects and their practical images on display subjected by the LiDAR sensor employing a 905 nm wavelength.

**Figure 2 nanomaterials-12-03689-f002:**
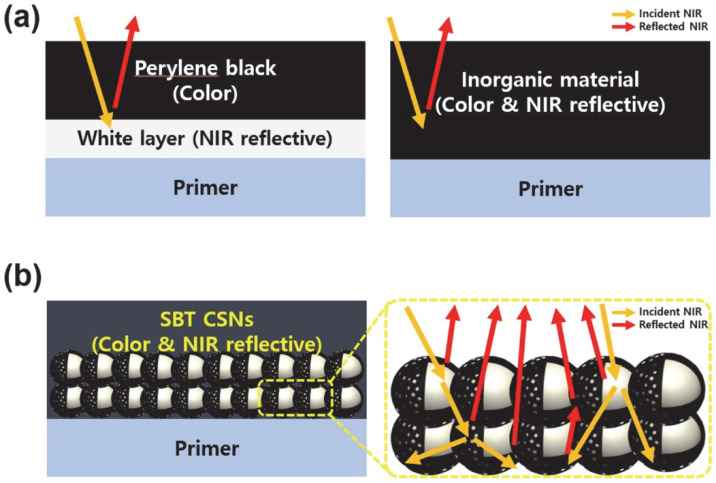
Schematic illustrations of NIR reflectance mechanisms of (**a**) previously reported bilayered (separate color/NIR reflective layer) organic perylene black with white surfacer system and monolayered (color + NIR reflective layer) heavy metal-based inorganic pigment and (**b**) black-colored, monolayered, and LiDAR-detectable SBT CSNs synthesized in this study without heavy metal inclusion.

**Figure 3 nanomaterials-12-03689-f003:**
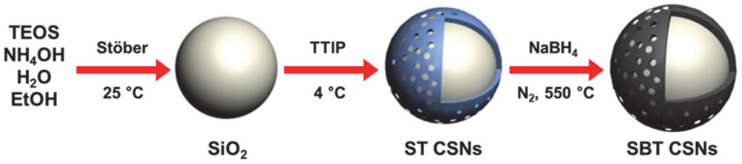
Schematic illustration for synthesis method of SiO_2_ nanoparticles, ST CSNs, and SBT CSNs via sequential Stöber, sol-gel, and NaBH_4_ reduction methods, respectively.

**Figure 4 nanomaterials-12-03689-f004:**
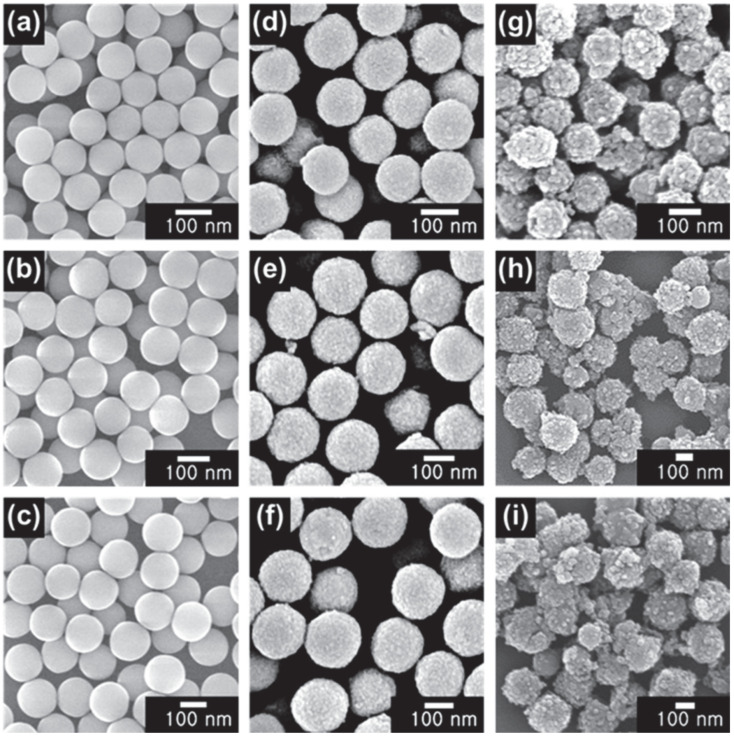
SEM images of (**a**) 100 nm SiO_2_, (**b**) 120 nm SiO_2_, (**c**) 140 nm SiO_2_, (**d**) 130 nm ST CSNs, (**e**) 160 nm ST CSNs, (**f**) 190 nm ST CSNs, (**g**) 140 nm SBT CSNs, (**h**) 170 nm SBT CSNs, and (**i**) 200 nm SBT CSNs.

**Figure 5 nanomaterials-12-03689-f005:**
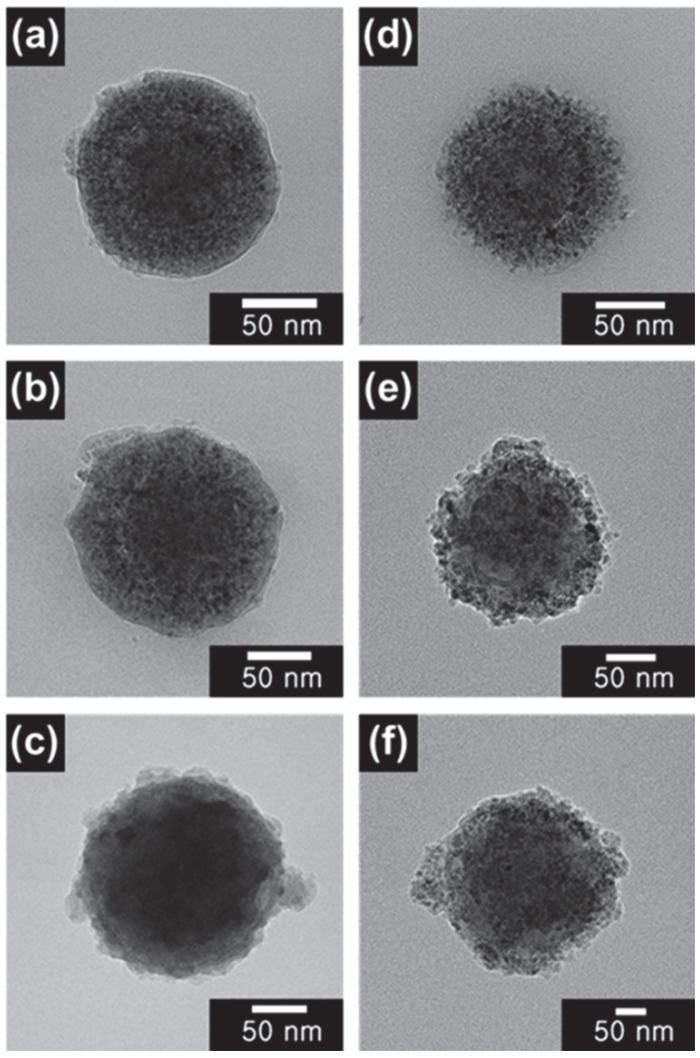
TEM images of (**a**) 130 nm ST CSNs, (**b**) 160 nm ST CSNs, (**c**) 190 nm ST CSNs, (**d**) 140 nm SBT CSNs, (**e**) 170 nm SBT CSNs, and (**f**) 200 nm SBT CSNs.

**Figure 6 nanomaterials-12-03689-f006:**
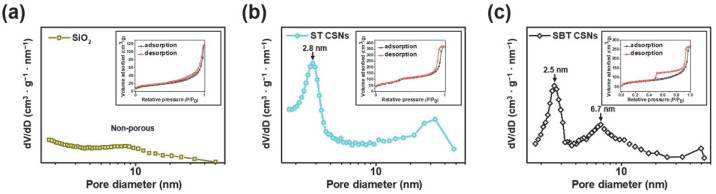
BJH pore size distribution curves of (**a**) 100 nm SiO_2_ nanoparticles, (**b**) 130 nm ST CSNs, and (**c**) 140 nm SBT CSNs (inset: N_2_-sorption isotherms of respective materials). For ST CSNs and SBT CSNs, small intrapores (*ca*. 2–6 nm) were created within TiO_2_ shells.

**Figure 7 nanomaterials-12-03689-f007:**
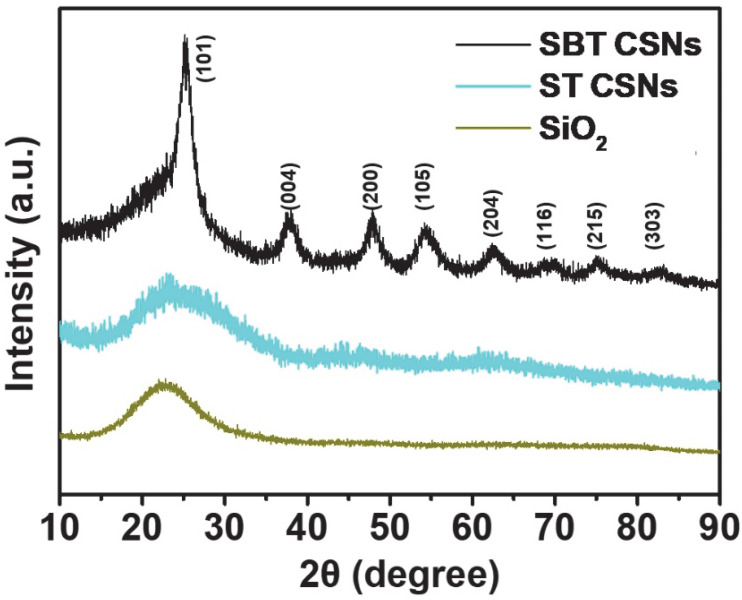
XRD diffraction patterns of 100 nm SiO_2_ nanoparticles, 130 nm ST CSNs, and 140 nm SBT CSNs.

**Figure 8 nanomaterials-12-03689-f008:**
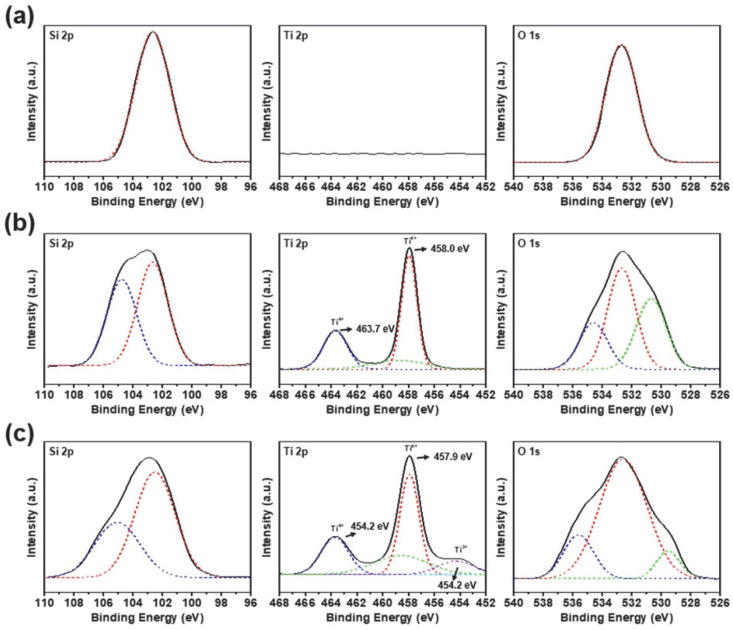
XPS Si 2p, Ti 2p, and O 1s level spectra of (**a**) 100 nm SiO_2_ nanoparticles, (**b**) 130 nm ST CSNs, and (**c**) 140 nm SBT CSNs.

**Figure 9 nanomaterials-12-03689-f009:**
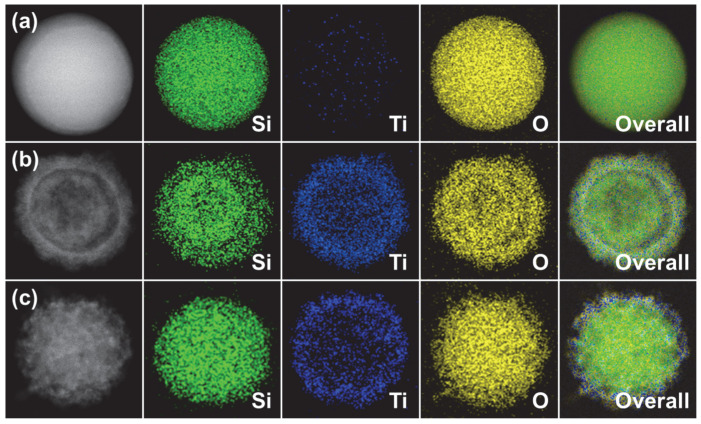
HAADF-STEM images and the corresponding elemental mapping images of (**a**) 100 nm SiO_2_ nanoparticles, (**b**) 130 nm ST CSNs, and (**c**) 140 nm SBT CSNs displaying the presence of Si (green), Ti (blue), and O (yellow) elements. Overall images were acquired by superimposing HADDF and each elemental image.

**Figure 10 nanomaterials-12-03689-f010:**
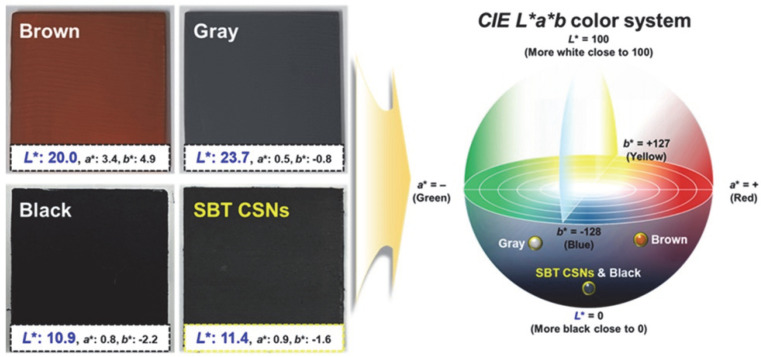
Digital photographs of SBT CSN-based paint and black, brown, and gray paints applied on glass substrates and their color descriptions in CIE *L*a*b** system. Black and SBT CSN-based paints displayed comparable blackness (*L** < 20).

**Figure 11 nanomaterials-12-03689-f011:**
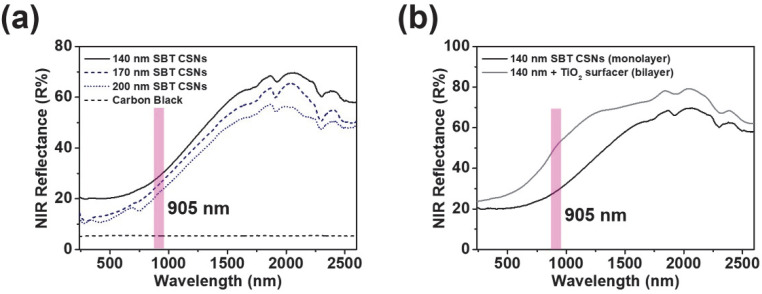
UV-VIS-NIR reflection spectra of (**a**) single coatings of various-sized SBT CSN- and carbon black-based paints on glass substrates and (**b**) 140 nm SBT CSN-based paint monolayer and bilayered system consisting of 140 nm SBT CSN-based paint and TiO_2_ surfacer (pink mark: 905 nm wavelength region).

**Figure 12 nanomaterials-12-03689-f012:**
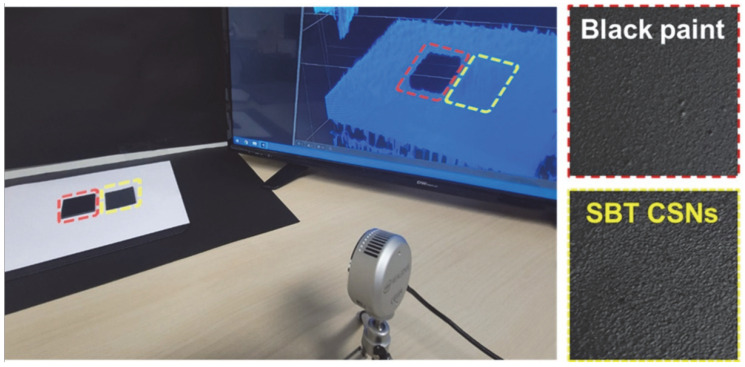
Digital photograph of 140 nm SBT CSN-based and black paints (monolayer on glass substrate) and their practical images on display by LiDAR sensor employing a 905 nm wavelength (LiDAR operation in depth mode).

**Table 1 nanomaterials-12-03689-t001:** BET specific surface areas and pore volumes of various-sized SiO_2_ nanoparticles, ST CSNs, and SBT CSNs.

	BET Specific Surface Area (m^2^∙g^−1^) *^a^*	Pore Size (nm)	Pore Volume (cm^3^∙g^−1^) *^b^*
100 nm SiO_2_	74.1 m^2^∙g^−1^	-	0.28 cm^3^∙g^−1^
120 nm SiO_2_	63.0 m^2^∙g^−1^	-	0.25 cm^3^∙g^−1^
140 nm SiO_2_	45.6 m^2^∙g^−1^	-	0.23 cm^3^∙g^−1^
130 nm ST CSNs	262.2 m^2^∙g^−1^	2.8 nm	0.56 cm^3^∙g^−1^
160 nm ST CSNs	210.7 m^2^∙g^−1^	2.9 nm	0.48 cm^3^∙g^−1^
190 nm ST CSNs	198.2 m^2^∙g^−1^	2.8 nm	0.43 cm^3^∙g^−1^
140 nm SBT CSNs	208.8 m^2^∙g^−1^	2.5 and 6.7 nm	0.38 cm^3^∙g^−1^
170 nm SBT CSNs	178.2 m^2^∙g^−1^	2.6 and 7.0 nm	0.32 cm^3^∙g^−1^
200 nm SBT CSNs	150.3 m^2^∙g^−1^	2.6 and 7.1 nm	0.29 cm^3^∙g^−1^

*^a^* Calculated by BET method. *^b^* Total pore volume.

**Table 2 nanomaterials-12-03689-t002:** Ratios of Ti species of various-sized ST CSNs and SBT CSNs as determined from their Ti 2p spectra.

Material	Ratio of Ti Species *^a^*	
	Ti^4+^	Ti^3+^
130 nm ST CSNs	95	5
160 nm ST CSNs	96	4
190 nm ST CSNs	94	6
140 nm SBT CSNs	83	17
170 nm SBT CSNs	81	19
200 nm SBT CSNs	77	23

*^a^* Ratio of Ti species as calculated by dividing the number of specific Ti species by the total number of species.

**Table 3 nanomaterials-12-03689-t003:** NIR reflectances of SBT CSN-based paint in monolayered and bilayered systems at 905 nm *^a^*.

Base Material (Layer System)	Reflectance at 905 nm (R%)
140 nm SBT CSNs (monolayer)	28.2
170 nm SBT CSNs (monolayer)	25.2
200 nm SBT CSNs (monolayer)	22.1
140 nm SBT CSNs/TiO_2_ surfacer (bilayer)	50.2
Commercial carbon black (black, monolayer)	5.4

*^a^* Materials were coated on glass substrates.

**Table 4 nanomaterials-12-03689-t004:** Comparison of NIR properties of SBT CSNs with other dark-tone LiDAR-detectable materials.

Materials	NIR Reflectance	CIE *L*a*b** System *^a^*			Layer System	Drawbacks	Ref
		*L**	*a**	*b**			
SBT CSNs *^b^*	28.2 R% (905 nm)	11.4	0.9	−1.6	Monolayer	–	Present work
Carbon black	5.4 R% (905 nm)	7.2	0.96	1.12	–	–	–
Mg-doped ZnFeO_4_	51 R% (700–2500 nm)	49.01	14.58	9.68	Monolayer	High *L****** value	[9]
Al_2_O_3_-doped CoFe_2_O_4_	24.5 R% (700–2500 nm)	23.09	0.66	−0.88	Monolayer	Low NIR reflectance	[10]
Ce_1−x_Gd_x_VO_4_	66.3 R% (700–2500 nm)	22.2	5.71	2.77	Monolayer	Heavy metal included	[11]
Perylene-derived	67.71 R% (905 nm)	28.88	4.6	−1.1	Bilayer	Low productivity	[12]

*^a^* Standard for true black color in the *L*a*b** system is *L** < 20 with *a** and *b** in the range of −3 to 3. *^b^* For the bilayer system, NIR reflectance of SBT CSNs was determined to be 50.2 R%.

## Data Availability

Data are contained within the article.

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
