# Peer review of "Synthesis of LiDAR-Detectable True Black Core/Shell Nanomaterial and Its Practical Use in LiDAR Applications"

_nanomaterials, 2022, doi:10.3390/nano12203689_

Round 1
Reviewer 1 Report
In this manuscript, the author reported a newly designed SiO2/black TiO2 core/shell nanoparticles (SBT CSNs), which are a highly LiDAR-detectable material. The SBT CSNs were synthesized using a combination of the sol-gel and NaBH4 reduction methods and exhibited a very high NIR reflectance, which are approximately 28.2% and 50.2% in the monolayer (without a TiO2 surfacer) and bilayer (with TiO2 surfacer) forms, respectively. In my opinions, it’s a very importance work, which could promote the development of LiDAR-detectable material. Thus, I would recommend it to be published after minor revision. The comments are as follows:
1. The main focus of the article is about a new SiO2/black TiO2 core/shell nanoparticles using LiDAR-detectable material. But at the same time, some corresponding LiDAR-detectable material performance should be listed and analyzed to make the article more logical and convincing.
2. The format of the reference documents is not uniform. There are also many problems with the format of reference documents, and some documents do not indicate page numbers. The format should be unified. Such as ref. 22.
3. The Figure 6 and Figure 10 are supposed to rearrange. Horizontal layout is better than vertical layout.
4. Pay attention to the format of the article picture. Figure 5 and subtitle are separated.
5. The English writing should be improved. There are some grammar problems existing in manuscript. Even though it's a boring job, it could make your work perfect and publish early.
Author Response
Response to Reviewer 1’s comments
Dear Reviewer
Thank you for reviewing our manuscript. We have substantially revised to reflect the reviewer’s comments on the manuscript. The following revisions were made on the basis of your comments.
---------------------------------------------------------------------------------------------------------
Reviewer 1
Comment) In this manuscript, the author reported a newly designed SiO2/black TiO2 core/shell nanoparticles (SBT CSNs), which are a highly LiDAR-detectable material. The SBT CSNs were synthesized using a combination of the sol-gel and NaBH4 reduction methods and exhibited a very high NIR reflectance, which are approximately 28.2% and 50.2% in the monolayer (without a TiO2 surfacer) and bilayer (with TiO2 surfacer) forms, respectively. In my opinions, it’s a very importance work, which could promote the development of LiDAR-detectable material. Thus, I would recommend it to be published after minor revision. The comments are as follows:
Response:
First of all, thank you for reviewing our manuscript. We’re grateful to you for the critical and valuable comments. We substantially revised the manuscript to take account of referee’s comments.
Comment 1) The main focus of the article is about a new SiO2/black TiO2 core/shell nanoparticles using LiDAR-detectable material. But at the same time, some corresponding LiDAR-detectable material performance should be listed and analyzed to make the article more logical and convincing.
Response:
Thank you for your valuable and helpful comment on this manuscript. To account for the suggestion of referee, we have listed the comparison of SBT CSNs with previously reported dark-tone LiDAR-detectable materials, as shown in Table R1. As-mentioned in the original manuscript, important standards for dark-tone LiDAR-detectable materials are known as blackness (L*), NIR reflectance at 905 nm (R%), layer system (mono or bilayer), and ecofriendliness (inclusion of heavy metal). For the comparison purpose, five previously-reported materials are evaluated with important standards (not many dark-tone LiDAR detectable materials are reported yet). Firstly, carbon black is known as the true black materials with L* value of 7.2, which can provide standard for blackness. In case of various inorganic LiDAR-detectable materials, each materials did not match the pre-mentioned standards. In specific, Mg-doped ZnFeO4 manifest high L* value of 49.01, far apart from the dark-tone or blackness [1]. For Al2O3-doped CoFe2O4, reported low NIR reflectance of 24.5 R% [2]. Also, Ce1–xGdxVO4 materials are having L* value of 22.2, which is not black color [3]. Furthermore, all of inorganic materials include heavy metals like Zn, Co, and Gd, which is not suitable for ecofriendly paint. Moreover, NIR reflectance of all inorganic materials not measured at specific 905 nm wavelength, which is common standard for commercialized LiDAR sensors. In case of perylene-derived organic materials, LiDAR detective layer system is composed as bilayer; color and NIR reflective white surfacer [4]. Therefore, low productivity and complicated coating process are known as the major drawbacks of perylene-derived LiDAR-detectable materials.
In this regard, we are humbly stating the novelty of newly developed SBT CSNs as suitable dark-tone LiDAR-detectable material with meeting the standards by including true blackness of L* = 11.4, high NIR reflectance as monolayer R% = 28.2 (bilayer: R% = 50.2 R%), and without usage of heavy metals. In the revised manuscript, we have strengthened the comparison table of SBT CSNs with previously reported dark-tone LiDAR-detectable materials and relevant explanation, as referee commented. Again, thank you for your valuable comment on our manuscript.
Table R1. Comparison of NIR properties of SBT CSNs with other dark-tone LiDAR-detectable materials.
Materials |
NIR reflectance |
CIE L*a*b* systema |
Layer system |
Drawbacks |
Ref |
||
L* |
a* |
b* |
|||||
SBT CSNsb |
28.2 R% (905nm) |
11.4 |
0.9 |
–1.6 |
Monolayer |
– |
Present work |
Carbon black |
5.4 R% (905nm) |
7.2 |
0.96 |
1.12 |
– |
– |
– |
Mg-doped ZnFeO4 |
51 R% (700–2500 nm) |
49.01 |
14.58 |
9.68 |
Monolayer |
High L* value |
[9] |
Al2O3-doped CoFe2O4 |
24.5 R% (700–2500 nm) |
23.09 |
0.66 |
–0.88 |
Monolayer |
Low NIR reflectance |
[10] |
Ce1–xGdxVO4 |
66.3 R% (700–2500 nm) |
22.2 |
5.71 |
2.77 |
Monolayer |
Heavy metal included |
[11] |
Perylene-derived |
67.71 R% (905 nm) |
28.88 |
4.6 |
–1.1 |
Bilayer |
Low productivity |
[12] |
a Standard for true black color in the L*a*b* system is L* < 20 with a* and b* in range of –3 to 3.
b For the bilayer system, NIR reflectance of SBT CSNs are determined to be 50.2 R%.
Revised parts in the manuscript
1) Table R1 was inserted as Table 4 in the revised manuscript.
2) The following sentences were inserted in the revised manuscript:
(a) “Newly designed SBT CSNs were compared with other dark-tone LiDAR detectable materials (Table 4). Compared with previously reported dark-tone LiDAR-detectable materials, it was clearly observed that the SBT CSNs were synthesized with true blackness (L* of 11.4), high NIR reflectance in monolayer (R% of 28.2, and no heavy metal inclusion. With various advantages, the SBT CSNs can be promising candidate for application in future autonomous vehicle.” (Page 11, Line 355)
Comment 2) The format of the reference documents is not uniform. There are also many problems with the format of reference documents, and some documents do not indicate page numbers. The format should be unified. Such as ref. 22.
Response:
Thank you for your critical comment on this manuscript. We solely agreed on the referee’s comment regarding on the uniformity of reference format. To account for the suggestion of referee, we have closely checked problems like spelling, format, and page numbers. Corrected version of references are inserted in the revised manuscript. Again, thank you for your helpful comment and carefully reviewing our manuscript.
Revised parts in the manuscript
1) The following references were modified and inserted in the revised manuscript:
(a) “Van Brummelen, J.; O’Brien, M.; Gruyer, D.; Najjaran, H. Autonomous vehicle perception: The technology of today and tomorrow. Transp. Res. Part C Emerg. Technol. 2018, 89, 384–406.” (Page 14, Line 430, Reference [1])
(b) “Jeong, J.; Yoon, T.S.; Park, J.B. Towards a Meaningful 3D Map Using a 3D Lidar and a Camera. Sensors 2018, 18, 2571.” (Page 14, Line 435, Reference [4])
(c) “Jang, Y.; Yoon, C.-M.; Kim, S.; Lee, I.; Jang, J. Single/Dual Alkaline Earth Metal-Doped Hollow Nanoparticles as Nanocarrier for Accelerating Neurite Development by Activating PERK and PJNK. Part. Part. Syst. Charact. 2018, 35, 1800132.” (Page 15, Line 482, Reference [27])
(d) “Yoon, C.-M.; Lee, S.; Cheong, O.J.; Jang, J. Enhanced Electroresponse of Alkaline Earth Metal-Doped Silica/Titania Spheres by Synergetic Effect of Dispersion Stability and Dielectric Property. ACS Appl. Mater. Inter. 2015, 7, 18977–18984.” (Page 15, Line 484, Reference [28])
(e) “Bhambhani, M.R.; Cutting, P.A.; Sing, K.S.W.; Turk, D.H. Analysis of Nitrogen Adsorption Isotherms on Porous and Nonporous Silicas by the BET and As Methods. J. Colloid Interface Sci. 1972, 38, 109–117.” (Page 15, Line 486, Reference [29])
(f) “Van De Krol, R.; Goossens, A.; Meulenkamp, E.A. In Situ X-Ray Diffraction of Lithium Intercalation in Nanostructured and Thin Film Anatase TiO2. J. Electrochem. Soc. 1999, 146, 3150–3154.” (Page 15, Line 499, Reference [35])
(g) “Junior, A.G.; Pereira, A.; Gomes, M.; Fraga, M.; Pessoa, R.; Leite, D.; Petraconi, G.; Nogueira, A.; Wender, H.; Miyakawa, W.; Massi, M.; Sobrinho, A.D.S. Black TiO2 Thin Films Production Using Hollow Cathode Hydrogen Plasma Treatment: Synthesis, Material Characteristics and Photocatalytic Activity. Catalysts 2020, 10, 282.” (Page 15, Line 506, Reference [38])
(h) “Marquez, A.; Serratosa, M.P.; Merida, J. Anthocyanin evolution and color changes in red grapes during their chamber drying. J. Agr. Food. Chem. 2013, 61, 9908-9914.” (Page 15, Line 513, Reference [42])
Comment 3) The Figure 6 and Figure 10 are supposed to rearrange. Horizontal layout is better than vertical layout.
Response:
Thank you for your valuable comment on our manuscript. We completely agreed on referee’s comment regarding on the rearrangement of Figure. As referee commented, we have rearranged the Figure 6 (BJH pore size distribution) and Figure 10 (UV-VIS-NIR reflection spectra) of original manuscript in horizontal layout, as shown in Figure R1 and R2, respectively. In the revised manuscript, rearranged figures are inserted in Results and Discussion section. Again, thank you for your valuable comment on out manuscript.
Figure R1. BJH pore size distribution curves of (a) 100 nm SiO2 nanoparticles, (b) 130 nm ST CSNs, and (c) 140 nm SBT CSNs [inset: N2-sorption isotherms of respective materials].
Figure R2. UV-VIS-NIR reflection spectra of (a) single coatings of various-sized SBT CSNs- and carbon black-based paints on glass substrates and (b) 140 nm SBT CSNs-based paint monolayer and bilayered system consisting of 140 nm SBT CSNs-based paint and TiO2 surfacer.
Revised parts in the manuscript
1) Figure R1 was inserted as Figure 6 in the revised manuscript.
2) Figure R2 was inserted as Figure 10 in the revised manuscript.
Comment 4) Pay attention to the format of the article picture. Figure 5 and subtitle are separated.
Response:
Thank you for your valuable comment on this manuscript. Firstly, we are sorry for confusion that we have made by separating figure and subtitle. To account for the suggestion of referee, we have corrected the format of Figure 5 and its subtitle of original manuscript, as shown in Figure R3. Again, thank you for your helpful comment on our manuscript.
Figure R3. TEM images of (a) 130 nm ST CSNs, (b) 160 nm ST CSNs, (c) 190 nm ST CSNs, (d) 140 nm SBT CSNs, (e) 170 nm SBT CSNs, and (f) 200 nm SBT CSNs.
Revised parts in the manuscript
1) Figure R3 and related subtitle were inserted as Figure 5 in the revised manuscript.
Comment 5) The English writing should be improved. There are some grammar problems existing in manuscript. Even though it's a boring job, it could make your work perfect and publish early.
Response:
Thank you for your valuable comment on our manuscript. We solely agreed on the comment regarding on the grammar problems in the manuscript. To account for the suggestion of referee, we have thoroughly reviewed our manuscript and revised the English writing for publication. In the revised manuscript, corrections were strengthened in Introduction, Materials and Methods, Results and Discussion, and Conclusions section.
Revised parts in the manuscript
1) The following sentences were corrected and inserted in the Introduction section:
(a) “To implement a completely autonomous driving system, various sensor systems, including radars, cameras, and light detection and ranging (LiDAR) sensors, which act as the eyes of the autonomous vehicle, are extensively studied by various research groups and companies [3].” (Page 1, Line 35)
(b) “Recently, LiDAR-detectable (i.e., NIR-reflective) dark-tone materials are receiving wide attention.” (Page 1, Line 54)
(c) “Titanium dioxide (TiO2) is one of the most effective NIR-reflective materials in the 780–2500 nm range due to its white color [14].” (Page 3, Line 77)
2) The following sentences were corrected and inserted in the Materials and Methods section:
(a) “The mixed powder was carefully transferred into an alumina boat, which was placed in a tube furnace and heated for 40 min at 550 °C at a heating rate of 5 °C/min in a nitrogen atmosphere.” (Page 4, Line 132)
3) The following sentences were corrected and inserted in the Results and Discussion section:
(a) “The strategy employed in this study to develop a new black, LiDAR-detectable material was to design a material that is free of a bilayer, such as that used in the case of conventional perylene-based black pigments, as well as harmful heavy metals but still exhibits high NIR reflectance and ecofriendliness.” (Page 5, Line 185)
(b) “Finally, practical LiDAR sensor application of SBT CSNs was evaluated by employing the monolayer of 140 nm SBT CSNs-based and black paints, as shown in Figure 11.” (Page 12, Line 376)
4) The following sentences were corrected and inserted in the Conclusions section:
(a) “Moreover, a bilayered system consisting of a paint based on the SBT CSNs and a TiO2 surfacer manifested an even higher NIR reflectance at 50.2 R%, owing to the additional reflectance induced by the the escaped NIR light.” (Page 13, Line 402)
References
- Liu, L.; Han, A.; Ye, M.; Feng, W. The Evaluation of Thermal Performance of Cool Coatings Colored with High Near-Infrared Reflective Nano-brown Inorganic Pigments: Magnesium Doped ZnFe2O4 Sol. Energy 2015, 113, 48–56.
- Sangwong, N.; Suwan, M.; Supothina, S. Effect of Calcination Temperature and Dolomite or Al2O3 Doping on Properties of NIR–Reflective CoFe2O4 Black Pigment. Today Proc. 2019, 17, 1595–1601.
- Moriomoto, T.; Oka, R.; Minagawa, K.; Masui, T. Novel Near-Infrared Reflective Black Inorganic Pigment Based on Cerium Vanadate. RSC 2022, 12, 16570–16575.
- Lim, T.; Bae, S.H.; Yu, S.H.; Baek, K.-Y.; Cho, S. Near-Infrared Reflective Dark-Tone Bilayer System for LiDAR-Based Autonomous Vehicles. Res. 2022, 30, 342–347.
-----------------------------------------------------------------------------------------------------
We have made a considerable effort to address all the concerns of the referees in the revised manuscript.
The revised parts were underlined and highlighted by blue color in the manuscript. We are grateful to referees for the valuable comments again.
We would like to ask your indulgence in considering our paper and highly welcome your valuable suggestions for its publication. Thank you for your consideration. I am looking forward to hearing from you sooner or later.
With all good wishes
Sincerely yours,
Prof. Chang-Min Yoon
Department of Chemical and
Biological Engineering
Hanbat National University
Daejeon, Korea 34158
Tel) 82-42-821-1528
Fax) 82-42-821-1593
E-mail) [email protected]

Reviewer 2 Report
This manuscript reports the synthesis of SiO2/black TiO2 core/shell nanoparticles, the characterization of their microstructure, composition, NIR-reflective properties and their use in LiDAR sensoring. The outcomes are of interest to the readers. However, a few minor revisions are needed before being accepted for publication. Below please find the comments in detail.
1. The abstract needs to be rewritten. The first 7 sentences (L17-L26) need to be condensed and the quantitative results for microstructure, composition and optical properties of SiO2/black TiO2 core/shell nanomaterials need to be added.
2.In the manuscript L161, “5x5 cm” should be “5x5 cm2”.
Author Response
Response to Reviewer 2’s comments
Dear Reviewer
Thank you for reviewing our manuscript. We have substantially revised to reflect the reviewer’s comments on the manuscript. The following revisions were made on the basis of your comments.
---------------------------------------------------------------------------------------------------------
Reviewer 2
Comment) This manuscript reports the synthesis of SiO2/black TiO2 core/shell nanoparticles, the characterization of their microstructure, composition, NIR-reflective properties and their use in LiDAR sensoring. The outcomes are of interest to the readers. However, a few minor revisions are needed before being accepted for publication. Below please find the comments in detail.
Response:
First of all, thank you for reviewing our manuscript. We’re grateful to you for the critical and valuable comments. We substantially revised the manuscript to take account of referee’s comments.
Comment 1) The abstract needs to be rewritten. The first 7 sentences (L17-L26) need to be condensed and the quantitative results for microstructure, composition and optical properties of SiO2/black TiO2 core/shell nanomaterials need to be added.
Response:
Thank you for your valuable comment on this manuscript. We solely agreed on the referee’s comment on condensation of initial sentences and the addition of quantitative results of SiO2/black TiO2 core/shell nanomaterials (SBT CSNs) in abstract. As referee pointed out, we have modified the abstract with addressing the quantitative results for microstructure, composition and optical properties of the SBT CSNs in the revised Abstract section. Again, thank you for your valuable comment on our manuscript.
Revised parts in the manuscript
1) The following paragraph was modified and inserted in the revised manuscript:
(a) “Light detection and ranging (LiDAR) sensor utilize the near-infrared (NIR) laser with a wavelength of 905 nm. However, LiDAR sensors are vulnerable in detecting black or dark-tone materials of light-absorbing properties. In this study, SiO2/black TiO2 core/shell nanoparticles (SBT CSNs) are designed as LiDAR-detectable black materials. The SBT CSNs with size of 140, 170, and 200 nm were fabricated by a series of Stöber, TTIP sol-gel, and modified NaBH4 reduction methods. These SBT CSNs are detectable by a LiDAR sensor and, owing to their core/shell structure with intrapores on the shell (ca. 2–6 nm), can effectively function as both color and NIR-reflective materials. Moreover, the LiDAR-detectable SBT CSNs exhibit high NIR reflectance (28.2 R%) in monolayer system and true blackness (L* < 20) along with ecofriendliness and hydrophilicity, making them highly suitable for use in autonomous vehicles.” (Page 1, Line 17)
Comment 2) In the manuscript L161, “5x5 cm” should be “5x5 cm2”.
Response:
Thank you for your valuable comment on our manuscript. We totally agree with referee’s comment on the unit. As referee pointed out, we have corrected the miswritten unit from “5x5 cm” to “5x5 cm2”, and corrected unit is modified in the revised manuscript. Again, thank you for your helpful comment on this manuscript.
Revised parts in the manuscript
1) The following unit of area was modified and inserted in the revised manuscript:
(a) “5 × 5 cm2” (Page 4, Line 156)
-----------------------------------------------------------------------------------------------------
We have made a considerable effort to address all the concerns of the referees in the revised manuscript.
The revised parts were underlined and highlighted by blue color in the manuscript. We are grateful to referees for the valuable comments again.
We would like to ask your indulgence in considering our paper and highly welcome your valuable suggestions for its publication. Thank you for your consideration. I am looking forward to hearing from you sooner or later.
With all good wishes
Sincerely yours,
Prof. Chang-Min Yoon
Department of Chemical and
Biological Engineering
Hanbat National University
Daejeon, Korea 34158
Tel) 82-42-821-1528
Fax) 82-42-821-1593
E-mail) [email protected]
